# Isoenzymes of the Flavonoid and Phenylpropanoid Pathways Show Organ-Specific Regulation during Apple Fruit Development

**DOI:** 10.3390/ijms241814353

**Published:** 2023-09-20

**Authors:** Paolo Baldi, Elisa Asquini, Giovanni Nicolussi Golo, Francesca Populin, Mirko Moser

**Affiliations:** Department of Genomics and Biology of Fruit Crops, Research and Innovation Centre, Fondazione Edmund Mach, San Michele all’Adige, 38098 Trento, Italy; elisa.asquini@fmach.it (E.A.); giovanni.nicolussigolo@fmach.it (G.N.G.); francesca.populin@fmach.it (F.P.); mirko.moser@fmach.it (M.M.)

**Keywords:** gene expression, homologous genes, organ-specific expression, developmental stages, fruit growth, apple

## Abstract

Elucidating the molecular mechanisms controlling fruit development is a primary target for the improvement of new apple (*Malus* × *domestica* Borkh.) cultivars. The first two weeks of development following pollination are crucial to determine fruit characteristics. During this period, a lot of changes take place in apple fruit, going from rapid cell division to the production of important metabolites. In this work, attention was focused on the phenylpropanoid and flavonoid pathways responsible for the production of numerous compounds contributing to fruit quality, such as flavonols, catechins, dihydrochalcones and anthocyanins. A total of 17 isoenzymes were identified, belonging to seven classes of the phenylpropanoid and flavonoid pathways that, despite showing more than 80% sequence identity, showed differential expression regulation during the first two weeks of apple fruit development. This feature seems to be quite common for most of the enzymes of both pathways. Differential regulation of isoenzymes was shown to be present in both ‘Golden Delicious’ and a wild relative (*Malus mandshurica*), even though differences were also present. Each isoenzyme showed a specific pattern of expression in the flower and fruit organs, suggesting that genes coding for enzymes with the same function may control different aspects of plant biology. Finally, promoter analysis was performed in order to highlight differences in the number and type of regulatory motifs. Overall, our results indicate that the control of the expression of genes involved in the phenylpropanoid and flavonoid pathways may be very complex as not only enzymes belonging to the same class, but even putative isoenzymes, can have different roles for the plant. Such genes may represent an important regulatory mechanism, as they would allow the plant to fine-tune the processing of metabolic intermediates towards different branches of the pathway, for example, in an organ-specific way.

## 1. Introduction

Apple (*Malus* × *domestica* Borkh.) is one of the most widely studied fruit trees worldwide. Due to its primary economic importance, a lot of efforts have been made in order to improve the quality of apple fruit [1]. There are numerous parameters that are important to determine apple fruit quality, such as size, color, sugar content, acidity and the accumulation of many different aromas [2]. All these characteristics are genetically controlled; therefore, it is important to elucidate the molecular mechanisms regulating apple fruit development.

Phenolic compounds, such as flavonols, flavanols, hydroxycinnanates and anthocyanins, are produced by apple fruit and play a crucial role to determine the final quality of the commercial product. Anthocyanins are responsible for the red coloration of apple skin and are strong antioxidants with a high nutraceutical value [3], while flavanols, especially proanthocyanidins, confer astringency to the fruit, a key parameter when cider apples are considered [4]. All phenolic compounds originate from phenylalanine through the phenylpropanoid pathway and the flavonoid pathway. The key enzymes of the phenylpropanoid pathway are phenylalanine ammonia lyase (PAL), cynnamate 4-hydroxylase (C4H) and 4-coumarate:CoA ligase (4CL), which synthesize p-coumaroyl-CoA starting from phenylalanine. Then p-coumaroyl-CoA can be channeled either to the flavonoid pathway or to the monolignol pathway, whose main product is lignin [5]. This is an important branching point, and in apple, an opposite regulation of the flavonoid and monolignol pathways was shown during the early stages of fruit development [6]. The first enzymes of the flavonoid pathway are chalcone synthase (CHS) and chalcone isomerase (CHI) synthesizing naringenin, that in turn can be used as a starting substrate to produce many different phenolic compounds, such as flavonols, isoflavonoids, anthocyanins and proanthocyanidins [5]. The biosynthetic regulation of the phenylpropanoid pathway and the flavonoid pathway is very complex, not only because of the great number of different compounds that characterize these secondary metabolisms, but also because they respond to numerous developmental and environmental stimuli, including UV light, pathogen infection and abiotic stresses, such as drought or cold [7]. Many different transcription factors (TFs) act synergistically to regulate phenylpropanoid metabolism, for example, R2R3-MYB, basic helix-turn-helix TFs and WD40 proteins that form together the so-called MBW complex that regulates the structural genes of lignin and flavonoid biosynthesis [8]. The MBW complex is regulated by upstream TFs, such as NAC, WRKY and bZIP TFs [9]. Post-transcriptional regulation also exists and is controlled by miRNA targeting both structural genes and regulators of these pathways [7]. In apple, due to the highly duplicated genome (ref), more than one gene can code for each of the enzymes involved in flavonoid biosynthesis, adding a level of complexity to the regulation of each pathway as not all the genes coding for the same enzyme may be regulated in the same way. In a previous work, it was found that in apple, at least two enzymes of the phenylpropanoid pathway (PAL and 4CL) present different isoforms showing opposite regulation during the early stages of fruit development [6]. Thus, a regulatory function was hypothesized for the different isoenzymes that could canalize metabolic intermediates towards specific branches of the pathway. Examples of differentially regulated members of the same gene family are not rare among plants [10,11], but in many cases, there can be a considerable variation in the sequence of the different members belonging to the same gene family [12]. Therefore, the gene function can be different, and in the case of plant enzymes, the substrate may also change. Much more challenging is to study genes with highly homologous sequences as it is very difficult to separate them by whole genome approaches, such as RNA-seq.

The goal of this study was to identify genes coding for putative isoenzymes with contrasting expression regulation and to find indications about their roles. For this purpose, an in silico approach was performed, starting from RNA-seq data to preliminary screen for differences among the selected isoenzymes. The expression behavior observed in the time-course samples collected in the first two weeks of fruit development was then validated through a quantitative real-time PCR (qPCR) approach. In addition, an organ-specific experiment was performed to better define the expression pattern of each gene. Possible different roles of isoenzymes belonging to the phenylpropanoid and flavonoid pathways were discussed.

## 2. Results

### 2.1. Sequence Selection and RNA-Seq Analysis

The initial gene set was selected manually from UniProt (https://www.uniprot.org accessed on 20 September 2019). Sequences of *Malus domestica* coding for enzymes belonging to the phenylpropanoid and flavonoid pathways were chosen as reference sequences. The corresponding nucleotide sequences were retrieved and used for a BLAST search at NCBI. All the apple genes showing more than 80% of identity at both the nucleotide and amino acid level with each reference sequence were then selected for further analysis. A total of 144 putative genes were found (Appendix A), representing our reference dataset for the RNA-seq analyses. 

The next step was to identify unequivocally each gene inside the RNA-seq apple libraries in order to evaluate the level of expression. The libraries were prepared at 0, 3, 7 and 14 days after anthesis (DAA) and yielded a number of reads between 23 million and 72.9 million paired-ends reads. After alignment and filter selection of the reads, it was possible to identify 41 unique transcripts (Appendix A) that were expressed in one or more libraries. The TPM value for each transcript was retrieved and used to visually estimate the level of expression of each gene at each time point. A total of 27 transcripts were selected, showing reads in all the time points, and then they were grouped according to their function (Figure 1). 

In total, eight groups were represented by two or more genes with the same putative function but showing differential regulation during the early stages of apple fruit development, precisely phenylalanine ammonia-lyase (PAL), trans-cynnamate 4-monoxigenase (C4H), 4-coumarate ligase (4CL), chalcone synthase (CHS), chalcone isomerase (CHI), dihydroflavonol 4-reductase (DFR), leucoanthocyanidin reductase (LAR) and anthocyanidin 3-O-glucosyltransferase (UFGT). In most cases, different members of the same gene family showed contrasting regulation (e.g., PAL, 4CL, DFR and others) with one or more genes that appeared to be upregulated or expressed at a high level during the entire period under analysis and others being downregulated during the same period. In some cases, a less clear situation was found (e.g., CHS) with all the genes of the same family showing more or less the same trend in expression but at different levels (Figure 1). An intermediate situation was found for genes such as UFGT and LAR (Figure 1). 

### 2.2. qPCR

The expression level estimated through the analysis of the RNA-seq reads aligning on each gene was based only on specific reads count at each time point without any statistical support. The idea was to manually identify and select candidates by visual comparison of the specific reads aligning to the targets at the different time points. Subsequently, to specifically analyze the expression levels of the selected transcripts, a quantitative real-time PCR (qPCR) experiment was performed at the same time points in order to confirm and validate these results.

All the eight gene families showing putative differences in gene expression according to the reads count were selected for qPCR experiments. For each family, highly specific primers were designed in order to distinguish the different isoforms (Table 1).

The clustering analysis and alignment performed on the groups selected in the expression coverage analysis allowed the visualization of regions with higher variability (Figure 2). 

In such regions, the primers were developed and tested to confirm their specificity for a single sequence. ‘Golden Delicious’ cDNA was used as a template for regular PCR, and the amplification product was loaded onto agarose gel. All primer pairs produced a single band that was eluted and sequenced. Sequencing results are shown in the Appendix A (Appendix A) and confirmed the specificity of all the primers tested. For each gene family, two or three genes were tested by qPCR. Expression results confirmed the presence of genes with contrasting regulation in most of the families, with the only exception of CHS (Appendix A). In detail, *MdPAL3* was shown to be induced during the entire period tested, reaching the maximum level, almost 20-fold higher than the control, at 14 DAA (Figure 3 and Appendix A). 

On the contrary, both *MdPAL1* and *MdPAL4* were found significantly repressed over all the time points (Figure 3 and Appendix A). A similar situation was found for CHI, with *MdCHI1* proving to be induced, while *MdCHI2* and *MdCHI3* were shown to be repressed (Figure 3 and Appendix A). For the LAR family, two genes were tested and showed opposite regulation with *MdLAR1* that was significantly induced during early fruit development, while *MdLAR2* was significantly repressed (Figure 3 and Appendix A). As for the 4CL family, *Md4CL4* was not significantly regulated during the early stages of apple fruit development, while *Md4CL1* was highly induced (Figure 3 and Appendix A). *Md4CL3* was slightly induced at 3 DAA and then repressed (Figure 3 and Appendix A). Two genes were tested for the UFGT family: the first (*MdUFGT1*) showed a clear induction over all the time points tested, while the second (*MdUFGT2*) showed a low increase in expression at 3 DAA, followed by a repression at 7 and 14 DAA (Figure 3 and Appendix A). Two gene families, DFR and C4H, showed less differences in gene expression. Two genes per family were tested and all were found upregulated, even though *MdDFR1* and *MdC4H1* showed a lower induction when compared to *MdDFR2* and *MdC4H2*, respectively (Figure 3 and Appendix A). In order to compare gene expression in different apple genotypes with contrasting fruit characteristics, a qPCR experiment was performed with the same primers also on Manchurian crabapple (*Malus mandshurica*). Overall, most of the gene families, such as PAL, LAR, CHI and C4H, showed comparable behavior in ‘Golden Delicious’ and Manchurian crabapple, with all the genes following the same trend in induction and repression (Appendix A). However, in some cases, significant differences in induction level were found, such as for *MdCHI1*, *MdPAL3*, *MdC4H1* and *MdC4H2*, with Manchurian crabapple showing a higher level of gene expression when compared to ‘Golden Delicious’ (Figure 4). For the DFR family, more pronounced differences were found between the two apple genotypes. *MdDFR1* showed a more rapid induction in ‘Golden Delicious’, reaching a peak at 3 DAA and then progressively decreasing until 14 DAA, while in Manchurian crabapple, a lower but steady induction was found during all the time points (Figure 4). On the contrary, *MdDFR2* expression was significantly higher in Manchurian crabapple, with a peak at 7 DAA, while ‘Golden Delicious’ showed a lower but constant induction (Figure 4). 

The genes belonging to the 4CL family were found differentially regulated in ‘Golden Delicious’ and Manchurian crabapple. *Md4CL1* appeared to be expressed only in ‘Golden Delicious’, while no detectable signal was found in Manchurian crabapple at any of the time points (Figure 4). *Md4CL4* was expressed at a low and constant level in ‘Golden Delicious’, while it was upregulated in Manchurian crabapple at 7 and 14 DAA (Figure 4). Finally, both *MdUFGT1* and *MdUFGT2* seemed to be upregulated only in ‘Golden Delicious’, while no detectable expression was found in Manchurian crabapple at any of the time points (Figure 4).

### 2.3. Organ-Specific Expression

In order to better define the expression pattern of each gene, an organ-specific study was performed by qPCR. Petals, anthers, ovary and sepals were analyzed separately in the case of flowers. In fruitlets, due to the very small size, only two different samples could be obtained: the external part, including the skin and part of the flesh (E), and the inner part, including the core and part of the flesh (I). The results confirmed the previous findings, giving at the same time a further hint regarding the complexity of gene regulation during the first stages of apple fruit development. For the PAL family, *MdPAL1* and *MdPAL4* showed a very similar expression pattern, with both genes expressed mostly in the petals and ovary, while a lower expression was found in the anthers and sepals. As expected from the time-course results, a very low expression was found in all the parts of the fruitlets (Figure 5). 

*MdPAL3* was induced in the fruitlets, especially in the external part, but also showed a significant expression in petals and sepals (Figure 5). For the C4H family, *MdC4H1* showed a general lower expression in all the considered tissues with respect to petals but a slightly higher expression in the external part of fruitlets at 14 DAA, though a considerable variation of the expression level among the different biological replicates characterized this analysis (Figure 5). *MdC4H2* showed far less variation among the biological replicates and was found to be highly induced mostly in the inner part of the fruitlets (Figure 5). For the 4CL family, all the three genes examined showed a distinct pattern of expression. As expected, both *Md4CL1* and *Md4CL4* were highly upregulated in fruitlets, compared to the petals, but Md4CL4 was also found to be expressed in the ovary and sepals (Figure 5). *Md4CL3* was expressed mostly in petals and was very low in all the other parts of the flower. Its expression decreased in fruitlets but remained higher in the external part than in the inner one (Figure 5). For the CHI family, *MdCHI1* showed an expression pattern similar to *Md4CH1*, though the high variation among the different biological replicates and the narrow range of difference in expression level between samples do not allow further considerations (Figure 6). 

On the contrary, *MdCHI2* and *MdCHI3* showed the same pattern, with both genes expressed exclusively in petals (Figure 6). As expected, both *MdDFR1* and *MdDFR2* were induced in fruitlets, with slightly different patterns as *MdDFR1* showed a decreased expression level at 14 DAA, while *MdDFR2* reached the highest expression level at the same time point (Figure 6). *MdLAR1* and *MdLAR2* showed quite simple and complementary patterns of expression, with *MdLAR1* being very highly expressed in all parts of fruitlets and *MdLAR2* in all parts of flowers (Figure 6). Finally, for the UFGT family, *MdUFGT1* was induced in the fruitlets, while *MdUFGT2* was expressed mostly in the petals but also with detectable expression in the anthers and sepals (Figure 6). 

### 2.4. Promoter Analysis

Promoter analysis was performed in order to find transcription factor-binding motifs and highlight differences in the number and type of such motifs in genes showing the same putative function and contrasting expression. The cDNA sequences were aligned against the apple genome and 2500 bp upstream the transcription start were analyzed. In two cases (*MdCHI2* and *MdCHI3*), it was not possible to retrieve sufficient sequence for the putative regulatory region as the best alignment of these genes was found downstream of a region with undefined sequence (Ns). For all the other genes, numerous putative transcription factor-binding motifs were found (Appendix A). In particular, 12 MYB and WRKY putative binding motifs, six light-responsive binding motifs, four abiotic-stress-related binding motifs and three biotic-stress-related binding motifs were selected for further analysis (Appendix A). All the promoter sequences were analyzed considering the number of each type of binding motif found in the considered regulatory region to create a motif-abundance profile (MAP) for each gene. The results are shown in Figure 7 in which genes with a similar MAP are clustered together. In most cases, genes with the same function and contrasting regulation also showed clear differences in their MAP and did not cluster together. 

This was true for PAL, UFGT and, with minor differences, for LAR gene families (Figure 7). At the same time, MdDFR1 and *MdDFR2*, which showed similar expression, also showed a similar MAP (Figure 7). Nevertheless, there were some exceptions: the most evident was the case of the 4CL family, in which *Md4CL1* and *Md4CL4* clustered together even if they showed contrasting gene expression (Figure 7). Finally, *MdC4H1* and *MdC4H2*, both showing induction of expression during the early stages of fruit development, were clearly separated according to their MAP (Figure 7).

## 3. Discussion

### 3.1. Gene Selection and RNA-Seq Analysis

Contrasting gene expression regulation within the same gene family is quite common in plants, especially when a great number of different genes belongs to the same family. Numerous cases are well documented for gene families with different roles, going from transcription factors [13] to mitogen-activated protein kinases [14] or enzymes [15]. In most cases, genes showing contrasting expression regulation also display clear differences in the nucleotide sequence and/or in the promoter sequence that may suggest their involvement in different processes [13]. In the case of genes with very similar sequences, it is usually supposed that the biological function is the same. In a previous work, two genes that belonged to the phenylpropanoid pathway and coded for PAL that showed a strong identity (88.6% at the amino acid level) but contrasting expression regulation during the early stages of fruit development were found [6], thus suggesting a different biological role for the two isoenzymes. Considering that the phenylpropanoid pathway and the strictly correlated flavonoid pathway are involved in numerous biological processes in plants, such as pathogen defense, UV protection, and plant and fruit growth [16,17], it was hypothesized that the plant could exploit different isoforms of the same enzyme, specifically controlling their expression to regulate flavonoid production and canalize metabolic flux toward different pathways to regulate the biosynthesis of many compounds, such as lignin, anthocyanins and flavonols [6]. In order to test this hypothesis, the existence of very similar isoforms with differential regulation was looked for to verify if this was a common feature for enzymes belonging to the phenylpropanoid and flavonoid pathways. To this purpose, several *Malus domestica* genes coding for enzymes belonging to these pathways were selected as a reference set, and the highly homologous sequences were retrieved by BLAST. As we were interested in test genes that are likely to share the same biological function, only those sequences sharing more than 80% of identity with each reference at both the nucleotide and amino acid level were considered. Even though most of the studies on regulatory gene functions are focused on transcription factors [18,19,20], the study of structural gene families and the roles of the different members is also important to understand complex plant traits [21,22]. Given the high number of candidates found, a procedure exploiting RNA-seq data was adopted, trying to select only the transcripts with interesting behavior to be further analyzed for their expression levels. Filtering of the alignments with stringent parameters and calculation of the TPM values per transcript allowed the evaluation of the expression trend in each gene during the considered time course. Focusing only on the normalized read counts produced by the specific alignments of each gene sequence used as a reference, it was possible to visually analyze the behavior of each gene and isoform through the time-course dataset. This approach, given that it is not supported by statistical evaluation of the RNA-seq expression levels, represented a preliminary screening for interesting candidates, reducing the efforts necessary to study their expression through RT-qPCR.

### 3.2. Gene Expression

According to the reads count, there were at least eight enzymes belonging to the phenylpropanoid or flavonoid pathway that presented two or more isoforms with a different regulation during the early stages of fruit development. The qPCR results were in accordance with those obtained using the RNA-seq libraries in most cases, indicating that the existence of different isoforms of the same enzyme with contrasting regulation is a common feature for both phenylpropanoid and flavonoid pathways. The organ-specific expression study further confirmed this hypothesis. Even though in some cases, high standard errors did not allow the level of expression among different organs to be clearly distinguished, in most cases, highly specific expression patterns were found. *MdCHI2* and *MdCHI3* were a clear example, being expressed exclusively in the petals. *MdCHI1*, showing 84% and 83.57% nucleotide homology with *MdCHI2* and *MdCHI3*, respectively, was found to have a completely different expression pattern. Other genes showed flower-specific (e.g., *MdPAL1*; *MdPAL4*; *MdLAR2*; *MdUFGT2*) or fruit-specific (e.g., *MdDFR2*; *MdLAR1*; *Md4CL1*) expression. For most of the gene family tested, at least two clearly different expression patterns were found between highly homologous genes. Taken together, our results suggest that only some specific isoforms are actually important for the activation of the flavonoid and phenylpropanoid pathways during fruit development [23], while the others may be involved in different processes, such as flowering [24], pathogen defense [25] or UV protection [26]. The highly specific primers developed in the present study could be easily used to verify this hypothesis by checking the expression of each isoform in different conditions. 

Most of the enzymes belonging to the phenylpropanoid and flavonoid pathways showing a differential regulation for different isoforms seem to be at key points of the pathway. PAL is the first enzyme of the phenylpropanoid pathway, and in *Arabidopsis thaliana*, it is encoded by four genes with different transcription patterns, with only *PAL1* and *PAL2* showing a putative redundant function [10,27]. 4CL is located right before a main bifurcation of the pathway, leading either to lignin or to flavonoid production. These two pathways showed a coordinated expression during the early stages of apple fruit development, with the lignin pathway being downregulated while the flavonoid pathway was upregulated [6]. 4CL isoenzymes have been isolated in several plant species and are supposed to regulate the flux of metabolites toward the production of a number of compounds, such as monolignols, flavonoids and coumarins [28,29]. LAR is involved in the biosynthesis of catechins, an important group of apple antioxidants [30]. Differential regulation of two members of the LAR family was already found in apple when Liao et al. [31] studied the expression of *LAR1* and *LAR2* in fruit skin at enlargement and mature stages. UFGT catalyzes the final glycosylation reaction in flavonoid biosynthesis and was associated with anthocyanin accumulation in several plant species [32,33]. Differential regulation of several members of the UFGT family was found in mango when expression analysis was performed on three genes (*MiUFGT1*, *MiUFGT3* and *MiUFGT4*) in leaves, flowers, roots and fruits [34]. Overall, our results correlate well with the previous findings, suggesting that differential regulation of highly homologous genes seems to be a common feature for both the phenylpropanoid and flavonoid pathways and may play an important regulatory role. Interestingly, such a mechanism proved to also be common in very distant apple genotypes, as observed in the comparison with Manchurian crabapple. Nevertheless, some differences were evident: for some of the genes studied, the regulation of the single isoforms was very similar between ‘Golden Delicious’ and Manchurian crabapple, such as for the PAL and LAR families. In other cases, the same isoform was upregulated in ‘Golden Delicious’ and not in Manchurian crabapple or vice versa, such as for *MdDFR1*, *MdDFR2*, *MdC4H1* and *MdC4H2*. This indicates that the flavonoid metabolism regulation may be different between ‘Golden Delicious’ and Manchurian crabapple during the early stages of fruit development, possibly influencing the final characteristics of the fruit. These results are in agreement with our previous findings, showing different levels of flavonoid metabolites accumulation between ‘Golden Delicious’ and Manchurian crabapple [6]. For *MdUFGT1*, *MdUFGT2* and *Md4CL3*, gene expression was almost undetectable in Manchurian crabapple. Liao et al. [31] found the same situation when comparing LAR gene expression in different *Malus domestica* and crabapple genotypes. The most likely explanation for these findings is that there are some mutations in the genomic DNA of the homologous crabapple gene, so that the primers used for amplification cannot recognize the sequence. An alternative explanation for this is that the gene Is really expressed at extremely low levels during the entire period under analysis in Manchurian crabapple, therefore indicating once again a difference in the flavonoid metabolism regulation between wild and domesticated apple genotypes.

The fact that highly homologous genes may be involved in different biological process in the plant is of primary importance for dissecting the mechanism regulating flavonoid and phenylpropanoid pathways in apple, especially in the case of plant transformation and gene editing. In fact, even though genetic transformation has been available since 1989 [35], several problems still exist in its application, such as the risk of off-target effects [36]. As an example, in a recent RNAi-based study, the silencing of a phloretin-specific glycosyltransferase gene (*UGT88F1*) resulted in the downregulation of other glycosyltransferases, such as *UGT88F4* and *UGT88F6* [37,38]. In another study, gene silencing of *MdPGT1* by conventional transgenesis and CRISPR/Cas9 genome editing led to different results. In the first case, the knockdown plants showed a phloridzin reduction, together with stunted phenotype and altered leaf morphology, while in the second case, the reduction in phloridzin did not lead to a dwarf phenotype, and the leaves were undistinguishable from control plants [39]. All these results indicate that more precise tools are required in order to fully dissect gene function in apple. The knowledge of each gene coding for putative isoenzymes could greatly increase the precision of plant transformation and gene editing approaches in specifically targeting the right candidate. 

### 3.3. Promoter Analysis

The promoter analysis revealed that genes with the same putative function and contrasting gene expression regulation usually also show clear differences in motif-abundance profile (MAP), but this is not a general rule as exceptions were found. Moreover, it was not possible to identify any specific transcription factor-binding motif correlating for the induction or repression of genes with similar transcription regulation during the early stages of apple fruit development. This is not surprising considering the high number of putative regulatory motifs and all the possible interactions among them. Moreover, it is necessary to consider all the other epigenetic mechanisms that may regulate gene expression, for example, three-dimensional chromatin organization [40] or DNA methylation [41]. Nonetheless, differences have been highlighted in the promoter regions of most of the genes analyzed, both in the type of regulatory motifs and in the copy number of each motif. Therefore, it can be hypothesized that such differences may be actually responsible for differential expression regulation during the early stages of apple fruit development and in different tissues/organs. It is also possible that instead of a single or few regulatory motifs, it could rather be the balance among the numerous binding factors interacting with the promoter that finally determine the level of expression of every single gene in different plant tissues or during different stages of fruit development. 

## 4. Materials and Methods

### 4.1. Plant Material

Flowers and fruits used in this study were harvested from apple trees grown and maintained at the orchard ‘Giaroni’, belonging to the Fondazione Edmund Mach (latitude 46.181539°, longitude 11.119877°). Five-year-old ‘Golden Delicious’ and Manchurian crabapple (*Malus mandshurica*) trees grafted on M9 rootstock were used. Apple flowers were harvested at anthesis, when they were fully open. Fruits were harvested at 3, 7 and 14 days after anthesis (DAA), during the cell division stage. For organ-specific extraction, petals, anthers, ovary and sepals were harvested separately. As for the fruitlets, due to the very small size (4 to 10 mm diameter), only two different parts were sampled: the external part, including the skin and part of the flesh (E) and the inner part, including the core and part of the flesh (I). All the plant materials were immediately frozen in liquid nitrogen and stored at −80 °C until used. 

### 4.2. RNA Extraction

Total RNA was extracted from apple flowers and fruits using the Spectrum plant total RNA kit (Sigma-Aldrich, St. Louis, MO, USA), according to the manufacturer’s instructions, including the on-column DNase digestion step. In order to assess the integrity of the genetic material, 1 µL of RNA was loaded onto 1.5% agarose gel and visualized using the ChemiDoc XRS gel imaging system (Bio-Rad, Hercules, CA, USA). RNA was then quantified using a Nanodrop 8000 spectrophotometer (Thermo Fisher Scientific, Waltham, MA, USA). cDNA synthesis was performed with Superscript III reverse transcriptase (Thermo Fisher Scientific, Waltham, MA, USA), using 5 μg of total RNA and 1 μL of oligo(dT)20 (50 μM) as a primer. 

### 4.3. cDNA Libraries and RNA-Seq Analysis

The RNA-seq libraries were prepared from the flowers and fruits harvested in the time course, starting from 5 ug of total RNA that was treated with the Ribo-Zero Magnetic kit Plant leaf (#MRZPL116; Epicentre) to remove ribosomal RNA. A total of 50 ng of ribosomal depleted RNA was subjected to the Scriptseq V2 RNA-Seq Library Preparation Kit (#SSV21124; Epicentre) for the preparation of Illumina-compatible libraries, following the manufacturer’s instructions. The libraries were then sequenced on a HiSeq2000 Illumina platform using 100 bp single-read mode. The sequencing data were submitted to the NCBI data repository with the BioProject number: PRJNA909076. The transcript sequences selected for the analysis were used as a reference for the alignment of RNA-seq reads. Bowtie2 [42,43] was used for the alignment step with default parameters and -N = 0. The alignment output was then filtered with an own python script, selecting only the reads with an alignment flag NM:i:0. Furthermore, another filter on the MAPQ flag was introduced using two different thresholds, one with MAPQ > 0 and another with MAPQ ≥ 4. The two datasets of aligned reads were used to calculate the per nucleotide coverage expressed as transcripts per kilobase million (TPM) for each selected gene transcript. The coverage was reported graphically and visually analyzed.

Transcripts having a coverage profile changing during the time course were selected and grouped based on their annotated function.

### 4.4. qPCR

Expression levels of selected genes were tested by qRT-PCR, using the ViiA 7 real-time PCR system (Life Technologies, Carlsbad, CA, USA). Platinum SYBR Green qPCR SuperMix-UDG (Life Technologies, Carlsbad, CA, USA) was used as fluorescent dye. Primers were designed using the online software Primer3Plus (v 3.3.0) [44]. The real-time PCR reactions were carried out using 1 µL of diluted (1:10) cDNA and the following reaction conditions: 2 min at 50 °C, 2 min at 95 °C, followed by 40 cycles at 95 °C for 15 s and 60 °C for 30 s. A control sample without the template was included for each primer combination and a melt curve analysis was performed at the end of each reaction in order to exclude unspecific amplification. All experiments were carried out using three independent biological replicates, consisting each of a bulk of five to eight flowers or fruits harvested from a single plant in random positions. Ct values were calculated by the ViiA 7 software based on the Ct values obtained from three technical replicates per sample. After testing several different housekeeping genes available from the literature, the two most stably expressed during the time points in study proved to be Actin [45] and EF1α [46], which were used as multiple reference genes in all the experiments. Expression profiles were obtained using the comparative Ct method [47], taking into account the efficiency of each primer combination calculated by means of dilution curves. Statistical analysis of the results was performed by *t* test.

### 4.5. Promoter Analysis

The transcripts selected for the qPCR analysis, were blastn [48] against the *Malus *× *domestica* v3.0 and v1.0 assemblies (https://www.rosaceae.org accessed on 7 November 2022). The corresponding genomic regions were selected when qcovs of the aligned transcript was ≥74 and pident ≥ 85%. Manual filtering correction was applied to discard those hits that in blastn passed the filtering only with partial sequence (i.e., homology for the entire length but drop in homology for part of the transcript sequence). Regions upstream of the start codon with a range of −2500 bp were retrieved as putative regulatory regions. Selected promoter regions were clustered to reduce redundancy using cd-hit-est with -c = 0.95. The consensus sequence of each cluster was then subjected to the NewPLACE TFDB [49], retrieving the TF-binding information.

## 5. Conclusions

The results presented in this work indicate that not only enzymes belonging to the same class, but even isoenzymes, may have different functions for the plant. This seems to be quite a common feature for both phenylpropanoid and flavonoid pathways. Such a feature appeared to be conserved among two different apple species and may have a regulatory function, allowing the plant to canalize metabolic intermediates toward different branches of the pathway according to the developmental and environmental needs. The organ-specific expression patterns seemed to confirm this hypothesis, suggesting that some genes, such as *MdDFR2*, *MdLAR1*, *MdC4H2* and *Md4CL1*, are probably involved in fruit development, while others, for example, *MdPAL1*, *MdPAL4*, *MdLAR2*, *MdUFGT2*, *MdCHI2* and *MdCHI3*, seemed to have different functions, likely in floral biology. Further studies are necessary in order to fully dissect the molecular mechanism controlling expression regulation of the different homologous genes and the specific transcription factors involved. The possibility to distinguish highly homologous sequences with contrasting regulation may have a first practical application in increasing the precision of plant transformation and gene editing approaches.

## Figures and Tables

**Figure 1 ijms-24-14353-f001:**
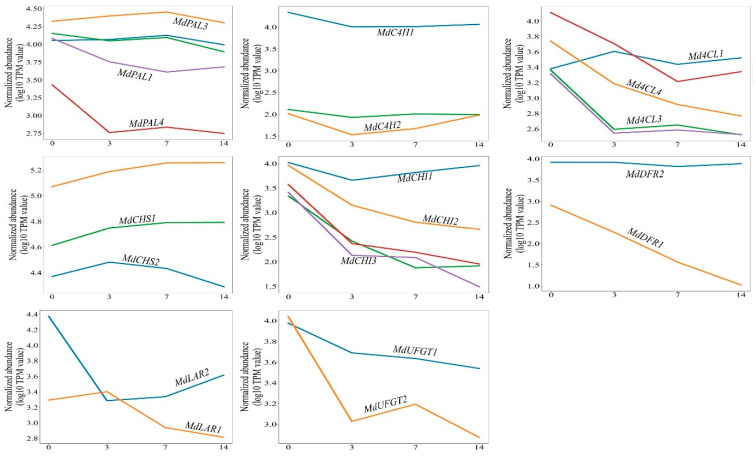
Expression level of the apple sequences showing reads in all the RNA-seq libraries. The colored lines represent the expression level of each gene estimated on the base of TPM values. The time points considered are indicated on the *x* axis and are expressed in DAA. On each panel the sequences selected for qPCR analysis are indicated with the corresponding gene name.

**Figure 2 ijms-24-14353-f002:**
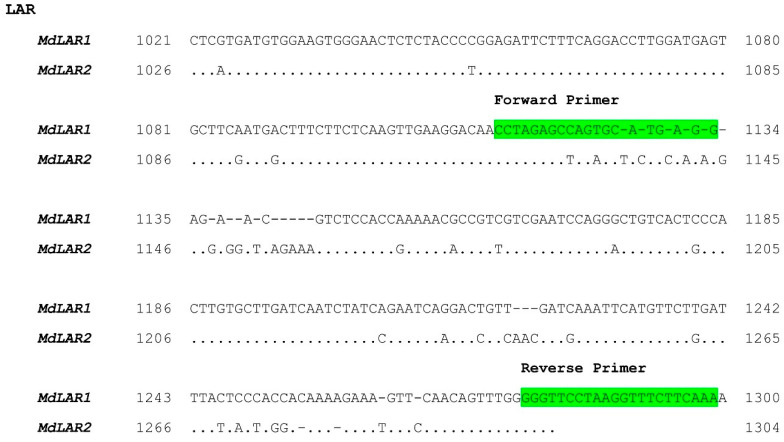
Alignment analysis and primer design. Alignment of *MdLAR1* and *MdLar2* is reported as an example. The specific primers were designed on the nucleotide regions with higher variability.

**Figure 3 ijms-24-14353-f003:**
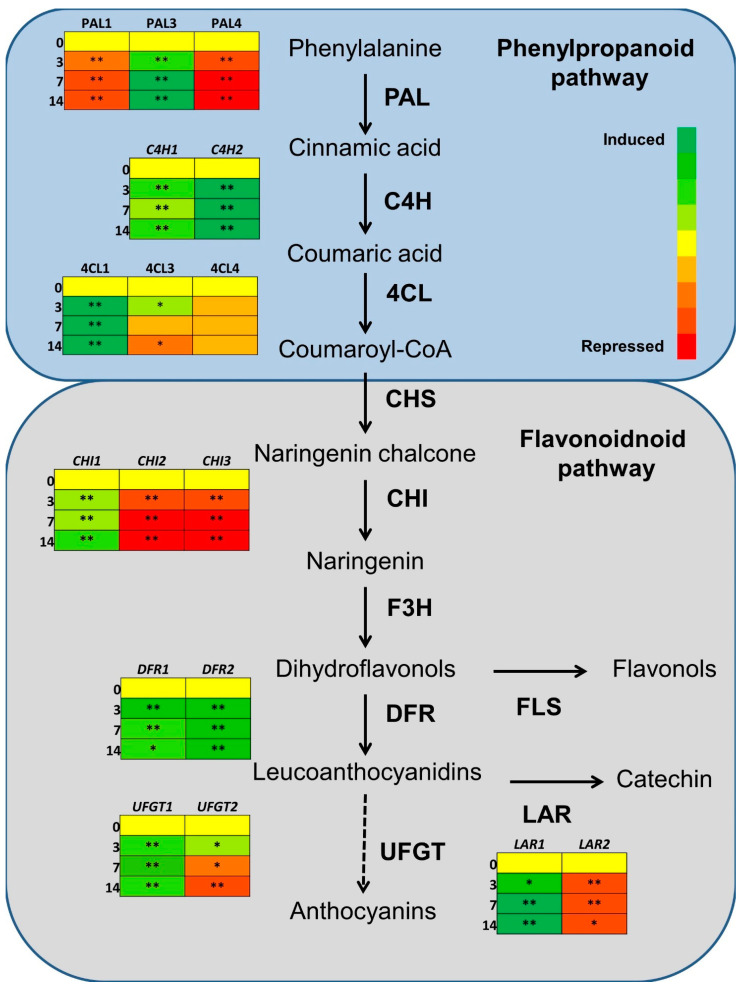
Schematic representation of phenylpropanoid and flavonoid pathways and expression levels of the apple genes studied at 0, 3, 7 and 14 DAA. Each panel shows the induction or repression of the corresponding gene according to the color scale shown on the right. Significant differences between each time point and the expression level at 0 DAA are indicated on each panel (* *p* < 0.05; ** *p* < 0.01). Abbreviations: PAL: phenylalanine ammonia lyase; C4H: cynnamate 4-hydroxylase; 4CL: 4-coumarate:CoA ligase; CHS: chalcone synthase; CHI: chalcone isomerase; F3H: flavanone 3-dioxygenase; DFR: dihydroflavonol 4-reductase; FLS flavonol synthase; UFGT: anthocyanidin 3-O-glucosyltransferase; LAR: leucoanthocyanidin reductase.

**Figure 4 ijms-24-14353-f004:**
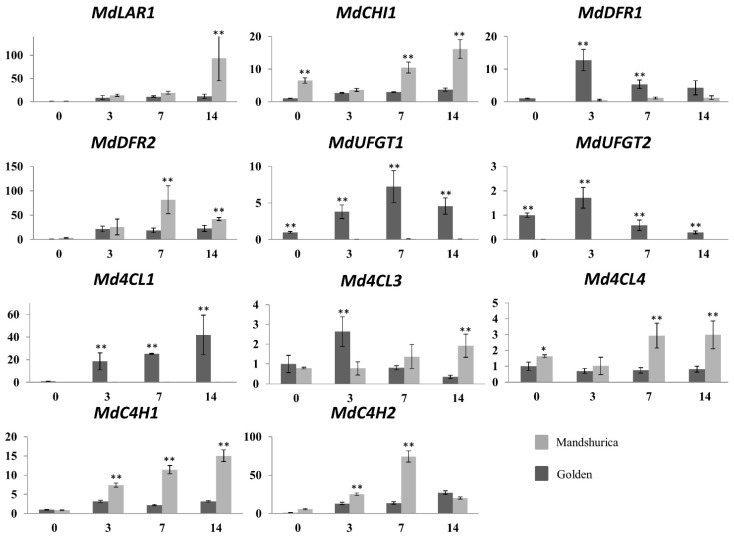
Relative expression levels of apple genes showing different trends in expression regulation in ‘Golden Delicious’ and Manchurian crabapple at 0, 3, 7, 14 DAA (*x* axis). On the *y* axis, the expression level relative to the value at 0 DAA in ‘Golden Delicious’ is indicated for each gene. All the data were normalized to actin and *EF1α*. Bars represent the standard deviation for three biological replicates. Significant differences between ‘Golden Delicious’ and Manchurian crabapple are indicated (* *p* < 0.05; ** *p* < 0.01).

**Figure 5 ijms-24-14353-f005:**
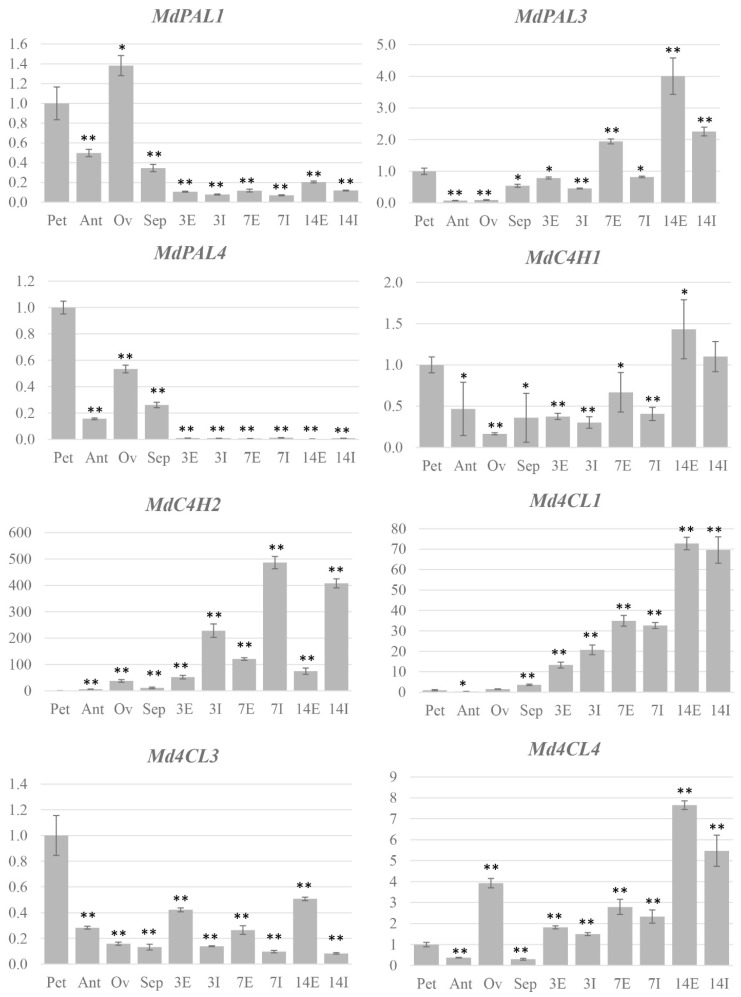
Relative expression level of apple genes belonging to the phenylpropanoid pathway in different organs of ‘Golden Delicious’. Petals (Pet); anthers (Ant); ovary (Ov); sepals (Sep); the numbers on the *x* axis indicate the DAA. On the *y* axis, the relative expression level of each gene is indicated, normalized against the value of petals. All the data were normalized to actin and *EF1α*. Bars represent the standard deviation for three biological replicates. Significant differences between means are indicated, referring to petals (* *p* < 0.05; ** *p* < 0.01). E: external part of the fruit, including the skin and part of the flesh; I: inner part of the fruit, including the core and part of the flesh.

**Figure 6 ijms-24-14353-f006:**
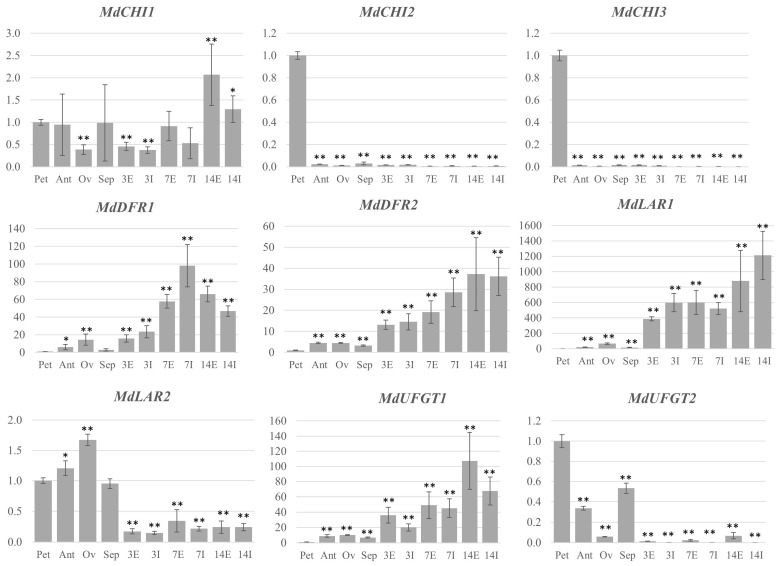
Relative expression level of apple genes belonging to the flavonoid pathway in different organs of ‘Golden Delicious’. Petals (Pet); anthers (Ant); ovary (Ov); sepals (Sep); the numbers on the *x* axis indicate the DAA. On the *y* axis, the relative expression level of each gene is indicated, normalized against the value of petals. All the data were normalized to actin and *EF1α*. Bars represent the standard deviation of three biological replicates. Significant differences between means are indicated, referred to petals (* *p* < 0.05; ** *p* < 0.01). E: external part of the fruit, including the skin and part of the flesh; I: inner part of the fruit, including the core and part of the flesh.

**Figure 7 ijms-24-14353-f007:**
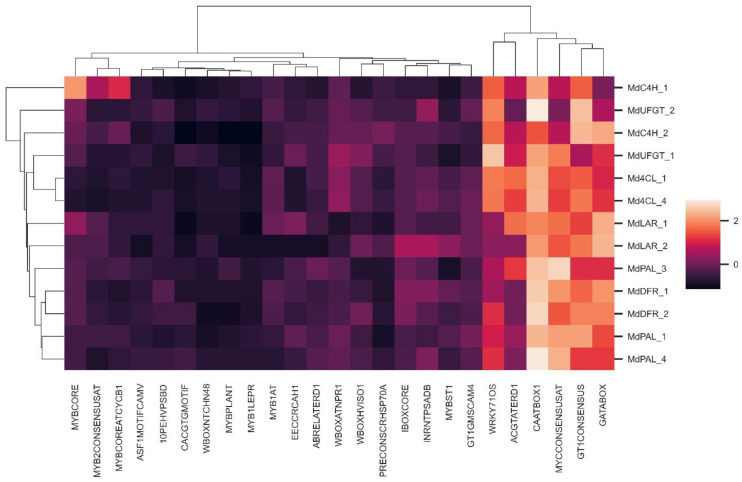
Heatmap of the matrix with the abundances of the selected motifs (columns) for the studied genes (rows). The rows were clustered based on their z-score and are considered as a motif-abundance profile (MAP) characteristic of each gene. The heatmap was generated using python seaborn.clustermap (https://doi.org/10.21105/joss.03021 accessed on 7 November 2022).

**Table 1 ijms-24-14353-t001:** List of specific primers used for qPCR analysis. The target sequence of each primer pair is also indicated.

Primer Name	Primer Sequence	Target Gene	ID
PAL_F1	GCTTTTGATTGGAAGGCCTAAC	*MdPAL1*	XM_008368428.3
PAL_R1	CTTCAGCGAAAATTGCCGAAAG		
PAL_F2	GGCTGCGATTGCCAACCATG	*MdPAL2*	XM_029105821.1
PAL_R2	TGTGAGGCAGAATATGGGTTG		
PAL_F3	CCGGAGAACAAACAAAGGAGC	*MdPAL3*	XM_008357397.3
PAL_R3	GAGGCGGTAATAGTACCGCG		
C4H_F1	CACGGGCTCCAACAAATGAG	*MdC4H1*	XM_029099060.1
C4H_R1	GGGAAGTTTCTGAACATCAGG		
C4H_F2	ATCTCTCCGACCTCGCCAG	*MdC4H2*	XM_029099564.1
C4H_R2	CTTGTTGGTGAAGAAAGGAAC		
4CL_F1	AAGCCCCCTCTTTCCAACAC	*Md4CL1*	XM_008364603.2
4CL_R1	CATGGTTAGTGGTGGCAGTG		
4CL_F4	GATTCATTTCTGCTAGCCTGC	*Md4CL4*	XM_029106270.1
4CL_R4	CATGGTTAATGGTGGCGGTG		
CHS_F1	AAGCCTTGTTTGGTGACGGC	*MdCHS1*	DQ026297.1
CHS_R1	CGTAATCAGACAGCACTTGTC		
CHS_F2	ACGTTTCACCTTTTGAAGGAC	*MdCHS2*	XM_029091251.1
CHS_R2	CCAGGCCCAAATCCAAATAG		
CHI_F1	AAGTTTCGGAGAATTGTGTATTC	*MdCHI1*	XM_008394013.3
CHI_R1	TTTACACCCGTTCAATAGTTGG		
CHI_F2	CATTGAAAAGTTCCTTGAGGTC	*MdCHI2*	XM_029100818.1
CHI_R2	CAGCACGATAAGATTCCTTGC		
CHI_F3	CAATGATACTGCCACTGACAG	*MdCHI3*	XM_029100826.1
CHI_R3	CAACAATTTAGATAACCTTGTGG		
DFR_F1	AGGAACCGTTATTATGGAAGAG	*MdDFR1*	XM_008379159.3
DFR_R1	GATTTAGTTCGTGTGATTGGTG		
DFR_F2	CTTTCGCCGATCTTAAGAAATG	*MdDFR2*	AF117268.1
DFR_R2	AACCAACTTTGACATCAACGAG		
LAR_F1	CCTAGAGCCAGTGCATGAGG	*MdLAR1*	AY830132.1
LAR_R1	TTTGAAGAAACCTTAGGAACCC		
LAR_F2	CATGTTCTGGATTTATTACTAGG	*MdLAR2*	AY830131.1
LAR_R2	TCCAACATTCACATCAAACTGG		
UFGT_F1	GGTCTCTCCAATGTACGAATC	*MdUFGT1*	AF117267.1
UFGT_R1	CCACTGTCCCGAAGCTTATG		
UFGT_F2	TGTACGTCAGTTTCGGGTCG	*MdUFGT2*	XM_008358841.3
UFGT_R2	TCCACATTTACCCCTATCTCC		

## Data Availability

The sequencing data were submitted to the NCBI data repository with the BioProject number: PRJNA909076.

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
