# Peer review of "Isoenzymes of the Flavonoid and Phenylpropanoid Pathways Show Organ-Specific Regulation during Apple Fruit Development"

_ijms, 2023, doi:10.3390/ijms241814353_

Round 1

Reviewer 1 Report

1. L9, L16: It should be defined more clearly for "the early developmental stages" and "the early stages".

2. L12: Avoid using first-person writing throughout the manuscript.

3. L14: "Numerous", but how many?

4. L15: "High sequence similarity", should be specified.

5. L19: "This mechanism seems to be conserved among species", but how do you know that?

6. L24-25: It's challenging to follow, particularly "may be more complex than expected". What do you expect at first and why?

7. L26-28: We already know that from the book of plant physiology, the point is to identify specific players and how they work in apple trees.

8. L80-86: It looks similar to several sentences in the abstract.

9. M&M: Add a section for statistical analysis.

10. L101, L111: What are the developmental stages of flowers and fruits?

11. Figures 5 and 6: Significant differences between means are all missing.

12. Figure 6: At least 7 standard errors are too high, e.g. MdCHI1-ant, sep, etc.

13. Conclusion: The authors should point out the most important findings of this study about possible molecular machinery and chief players, but not just give discussion, suggestions, and imagination.

Author Response

  1. L9, L16: It should be defined more clearly for "the early developmental stages" and "the early stages".

R: Sentences changed according to reviewer’s suggestion.

  1. L12: Avoid using first-person writing throughout the manuscript.

R: The manuscript was revised according to reviewer’s suggestion.

  1. L14: "Numerous", but how many?

R: Sentence modified specifying the number of enzymes studied.

  1. L15: "High sequence similarity", should be specified.

R: Sentence modified specifying the level of similarity.

  1. L19: "This mechanism seems to be conserved among species", but how do you know that?

R: Sentence changed.

  1. L24-25: It's challenging to follow, particularly "may be more complex than expected". What do you expect at first and why?

R: Sentence changed.

  1. L26-28: We already know that from the book of plant physiology, the point is to identify specific players and how they work in apple trees.

R: The novelty of the paper is that we showed that not only enzymes belonging to the same class may have different roles in plant physiology, but even putative isoenzymes. To our knowledge, the papers studying the different regulation of isoenzymes are few, especially in non-model plants. The text was changed to better explain this concept.

  1. L80-86: It looks similar to several sentences in the abstract.

R: Sentences changed.

  1. M&M: Add a section for statistical analysis.

R: The only statistical analysis performed was the t test (L486-487), therefore we believe that it is not necessary to create a specific section.

  1. L101, L111: What are the developmental stages of flowers and fruits?

R: Flowers were harvested at anthesis (L433); fruits were harvested at 3, 7 and 14 days after anthesis, during cell division stage (L433-434).

  1. Figures 5 and 6: Significant differences between means are all missing.

R: Significant differences are now indicated.

  1. Figure 6: At least 7 standard errors are too high, e.g. MdCHI1-ant, sep, etc.

R: It is true that in some cases the standard error is quite high, but in most cases the expression differences between the different organs can be clearly distinguished. A sentence in the Discussion section was added to assess that (L334-335).

  1. Conclusion: The authors should point out the most important findings of this study about possible molecular machinery and chief players, but not just give discussion, suggestions, and imagination.

R: The section was modified trying to be more specific about the findings illustrated in the paper.

Reviewer 2 Report

The authors investigated the phenylpropanoid and flavonoid pathways, which are responsible for the production of numerous compounds contributing to fruit quality, such as flavonols, catechins, dihydrochalcones, and anthocyanins, with numerous isoenzymes identified involved in these pathways. Furthermore, these enzymes showed differential expression regulations during the early stages of apple fruit development, whereas genes coding for enzymes with the same function may control different aspects of plant biology, consistent with the results of the promoter analysis based on different types of regulatory motifs.

I believe that the authors provided sufficient background, explained well the methodologies used, presented the data with appropriate figures and tables (some corrects are needed with the presentations of figures), and concluded appropriately based on available data. I have no major technical concerns, though I have seen some minor grammatical and editorial errors throughout the entire manuscript. Some of these errors are listed here for the authors to consider if a revision is requested by the editor.

Title: concise and clear

Abstract:

Line 18: be careful, parentheses should not be included inside another pair of parentheses.

Keywords: the authors may include “apple” in the keywords.

Introduction:

I believe that the authors have provided sufficient background.

Line 79: use “RNA-seq” instead of “RNAseq” throughout the entire manuscript.

Line 80: the authors should provide their explicit goals of this study somewhere in this paragraph.

M&M:

The explanations of the methodologies are sufficient for reproducibility.

Line 98: it is “–80 °C” but not “-80° C”

Results:

Both figures 1 and 4 should not be presented in lines but in bars (e.g., Figures 5 and 6), because lines may misleadingly indicate the values between the time points.

Figure 3: it is recommended that the authors should provide a list of abbreviations with full names given as well presented on this figure to help interpret this figure.

Figures 5 and 6:

The authors should indicate what the “I” and “E” mean in the numbers of DAF.

Figures 4-6:

The authors should explain well the units of Y-axis, especially these are relative expression levels.

Discussion:

Give the length of Discussion, I would highly suggest that the authors establish a few subsections to focus on the in-depth discussion of the topics of these subsections.

The language (English) is acceptable, though some minor grammatical and editorial errors need to be corrected.

Author Response

Abstract:

Line 18: be careful, parentheses should not be included inside another pair of parentheses.

R: The scientific name was abbreviated in order to eliminate a pair of parentheses. Alternatively, the complete name could be maintained enclosed in square brackets.

Keywords: the authors may include “apple” in the keywords.

R: Keyword added.

Introduction:

I believe that the authors have provided sufficient background.

Line 79: use “RNA-seq” instead of “RNAseq” throughout the entire manuscript.

R: The text was changed according to the reviewer’s suggestion.

Line 80: the authors should provide their explicit goals of this study somewhere in this paragraph.

R: The goal of the study is now clearly stated (L80-81).

M&M:

The explanations of the methodologies are sufficient for reproducibility.

Line 98: it is “–80 °C” but not “-80° C”

R: The text was changed.

Results:

Both figures 1 and 4 should not be presented in lines but in bars (e.g., Figures 5 and 6), because lines may misleadingly indicate the values between the time points.

R: Figures 4 and S4 were changed according to the reviewer’s suggestion. As for figure 1, in our opinion it would be much more difficult to read it if the results were presented in columns, especially in all those cases where more than two genes are present. Therefore, we propose to leave it as it is.

Figure 3: it is recommended that the authors should provide a list of abbreviations with full names given as well presented on this figure to help interpret this figure.

R: List of abbreviation added.

Figures 5 and 6:

The authors should indicate what the “I” and “E” mean in the numbers of DAF.

R: The figure captions were changed according to the reviewer’s suggestion.

Figures 4-6:

The authors should explain well the units of Y-axis, especially these are relative expression levels.

R: The figure captions were changed according to the reviewer’s suggestion.

Discussion:

Give the length of Discussion, I would highly suggest that the authors establish a few subsections to focus on the in-depth discussion of the topics of these subsections.

R: The Discussion is now divided in subsections.

Reviewer 3 Report

Reproductive biology, the study of flower organogenesis, the formation and maturation of fruits, is an actual and promising field of biology. Understanding the molecular mechanisms underlying these processes is important from both practical and theoretical points of view. For this reason, this experiment is a relevant and intended contribution to understanding the developmental characteristics of individual plants and has further prospects.

The applied methods are adequate and modern. The results are clear and objective. However, there are some minor remarks that need to be noted.

1. The Introduction section lacks references to literary sources (lines 50,54).

     2. In the Methods section, paragraph 2.1. the size of the fruits of the apple trees that were collected for analysis, the time of collection of the fruits, as well as the sample of plant material taken for research, were not indicated.

Author Response

  1. The Introduction section lacks references to literary sources (lines 50,54).

R: Reference added.

  1. In the Methods section, paragraph 2.1. the size of the fruits of the apple trees that were collected for analysis, the time of collection of the fruits, as well as the sample of plant material taken for research, were not indicated.

R: The size of the fruit was added (L436). Flowers were harvested at anthesis (L433); fruits were harvested at 3, 7 and 14 days after anthesis, during cell division stage (L433-434).

Round 2

Reviewer 1 Report

I don't have further questions.